# Genomic and Phenotypic Characterization of *Cutibacterium acnes* Bacteriophages Isolated from Acne Patients

**DOI:** 10.3390/antibiotics11081041

**Published:** 2022-08-02

**Authors:** Shukho Kim, Hyesoon Song, Jong Sook Jin, Weon Ju Lee, Jungmin Kim

**Affiliations:** 1Department of Microbiology, School of Medicine, Kyungpook National University, Daegu 41944, Korea; shukhokim@knu.ac.kr (S.K.); in75724@hanmail.net (J.S.J.); 2Avian Disease Division, Animal and Plant Quarantine Agency, Gimcheon 39660, Korea; bumb007@hanmail.net; 3Department of Dermatology, School of Medicine, Kyungpook National University, Daegu 41944, Korea; weonju@knu.ac.kr

**Keywords:** *Cutibacterium acnes*, bacteriophage, phage therapy, hydrophobicity

## Abstract

*Cutibacterium* *acnes* is a pathogen that can cause acne vulgaris, sarcoidosis, endodontic lesions, eye infections, prosthetic joint infections, and prostate cancer. Recently, bacteriophage (phage) therapy has been developed as an alternative to antibiotics. In this study, we attempted to isolate 15 phages specific to *C. acnes* from 64 clinical samples obtained from patients with acne vulgaris. Furthermore, we sequenced the genomes of these three phages. Bioinformatic analysis revealed that the capsid and tape measure proteins are strongly hydrophobic. To efficiently solubilize the phage particles, we measured the adsorption rate, one-step growth curve, and phage stability using an SMT^2^ buffer containing Tween 20. Here, we report the genotypic and phenotypic characteristics of the novel *C. acnes*-specific phages.

## 1. Introduction

*Cutibacterium acnes* (formerly *Propionibacterium acnes*) is a Gram-positive, rod-shaped bacterium that is aerotolerant and grows slowly under anaerobic conditions [1,2]. This bacterium is a major commensal bacterium residing on healthy human skin [3,4] and it is the etiological pathogen for human acne vulgaris [5,6]. It plays an important role in maintaining skin health, but it has also been implicated in the pathogenesis of several diseases and infections, including sarcoidosis, SAPHO (synovitis, acne, pustulosis, hyperostosis, and osteomyelitis) syndrome [7,8], endodontic lesions [9], eye infections [10], prosthetic joint infections [11,12], prostate cancer [13], and acne vulgaris (commonly called acne) [5,6].

Antibiotic-resistant bacteria pose a serious threat to human health and the economy. More than 23,000 deaths result from 2 million infections caused by antibiotic-resistant bacteria every year in the United States [14,15]. New alternatives to antibiotics are urgently required in the post-antibiotic era. Antibiotic resistance in *C. acnes* (briefly CA) is a major problem. Treatment of acne vulgaris has failed due to the development of antibiotic resistance [16,17]. From the 1980s to the 2000s, the antimicrobial resistance of CA increased by approximately 40% worldwide [17]. Resistance to erythromycin and/or clindamycin is predominant in the Americas, Europe, and Asia [18].

Bacteriophages (phages) can be used as an alternative to other antimicrobial agents, including antibiotics [19]. Phages can only multiply in an intracellular environment and they do so by infecting and killing the host bacterium. Recently, phage therapy has been developed as an alternative to antibiotics for the treatment of infections [20]. In particular, phages specific to CA can be used to treat CA infection. To be useful for phage therapy, the candidate phage should be lytic, non-lysogenic, and free of virulence factors and antibiotic resistance genes [21,22]. In this study, phages specific to CA were isolated from 64 clinical samples obtained from patients with acne vulgaris. The isolated phages were characterized genomically and phenotypically.

## 2. Results

### 2.1. Phage Isolation and Host Ranges

Fifteen phages were isolated from 64 samples propagated using the CA ATCC 11827 strain as the host bacterium. In the host spectrum assay, none of the 15 phages lysed *Staphylococcus aureus*, *S. epidermidis*, *Corynebacterium*, *Bacillus*, *Pseudomonas aeruginosa*, *Acinetobacter baumannii*, *Enterococcus faecium*, or *Escherichia coli*. In the host range determination assay, 16 clinical isolates of CA with eight different sequence types (STs) were exposed to the phage (Table 1). Phages P1, P10, and P14 lysed all 16 clinical isolates of CA. The other 12 phages were able to lyse almost all of the CA isolates, except two ST53 isolates (113 and 2878).

### 2.2. Morphological Analysis

Phage plaques were assessed using a CA bacterial lawn that had been cultured on a BHA plate. The plaques appeared as transparent halos of various sizes (1–10 mm) after the plates were cultured for 3 days at 37 °C under anaerobic conditions (Figure 1d). Transmission electron microscopy (TEM) of the 2012-15, P1, and P2 phages showed that the length of the phages was approximately 220 ± 20 nm (head to tail), the width of the head was 88 ± 5 nm, and that the phages had a non-contractile tail. The head was isometric and the tail appeared long and flexible (Figure 1a–c). These phages belonged to the *Siphoviridae* family. The three phages (2012-15, P1, and P2) exhibited similar morphologies. Phage aggregates were easily observed during TEM.

### 2.3. Genomic DNA Characterization of Three Phages, 2012-15, P1, and P2

Phage DNA was isolated from high titer (>10^10^ pfu/mL) phage solutions and the genomic DNA sequences were completely decoded. The genome sizes of the three phages were 29,741 bp (2012-15), 29,533 bp (P1), and 30,016 bp (P2). All of these genomes were similar in size to the average genome size of the CA phage (approximately 29,400 bp). The 2012-15, P1, and P2 genomes had 43, 43, and 4 open reading frames (ORFs), respectively. All three phage genomes had the same GC content (54% GC content). Interestingly, there were unusually low GC content regions (≤25%) between ORF1 and ORF2 in all three phage genomes (Figure 2).

Using BLAST for nucleotide sequence alignment and the 2012-15 phage genome as a query, we found 96 significantly aligned genome sequences. These were predominantly from *Propionibacterium* (*Cutibacterium*) phages, with more than 93% query coverage and 87.54% identity. Among the 96 phage genomes, the 2012-15 phage genome was the closest to the phage P2 genome and the farthest from the *Propionibacterium* phage Kubed (GenBank accession number KR337645) (Figure 3). The 2012-15, P1, P2, and Kubed phage genomes were also analyzed using the MAUVE program to identify any possible rearrangements of the conserved genomic sequences. The results of this analysis indicated that no rearrangements occurred among the four phages (Appendix A).

Genomic analysis using PhageLeads at www.phageleads.dk (accessed on 11 April 2022) did not suggest that genes enabling a temperate lifestyle, antimicrobial resistance, or virulence genes would be found in the genomes of the three phages.

### 2.4. Hydrophobic Characteristics of the Capsid and Tape Measure Proteins of the 2012-15 Phage

Interestingly, the head (capsid) proteins and tape measure proteins (TMPs) of the sequenced phages contained distinct proportions of hydrophobic amino acids. In particular, the capsid proteins and TMPs of these phages contained more hydrophobic amino acids than those of other phages (Table 2). The capsid proteins comprised 55.7% hydrophobic amino acids (AAs), 3.58% acidic AAs, 8.9% basic AAs, and 31.81% neutral AAs. Similarly, TMP comprised 50.77% hydrophobic AAs, 10.84% acidic AAs, 9.91% basic AAs, and 28.48% neutral AAs. The composition of the other structural proteins of the 2012-15 phage was different, with less than 50% hydrophobic AAs. Compared to the *E. coli* ADB-2 phage, whose capsid proteins and TMPs have 42.7% and 40.65% hydrophobic AAs, respectively, the capsid proteins and TMP of 2012-15 phage are more hydrophobic.

TMPs are conserved structural proteins in *Siphoviridae* phages that form channels through the host cell membrane. The phages use these channels to inject their genetic material into the host cell. The TMP of the 2012-15 phage was compared with the TMPs of 11 other phages (two *E. coli* phages, five *Lactococcus* phages, one *Bacillus* phage, one *Listeria* phage, one *Streptococcus* phage, and one *Staphylococcus* phage), and the TMP of the 2012-15 phage showed the highest hydrophobicity (Table 3). All ORFs and protein sequences of three phages are listed in Appendix A.

### 2.5. Adsorption Rate and One-Step Growth Curve of Phage 2012-15

The adsorption time and burst size of the 2012-15 phage were measured after optimizing the phage dilution buffer. One minute was required for at least 80% of the 2012-15 phage to be absorbed into CA host cells, which is the first stage of phage infection. Over 90% of the phages were absorbed after 3 min. Free phage levels remained at 5.63% and 3.50% at 10 and 20 min, respectively (Figure 4a).

*S. aureus* ATCC25923 was also used in the 2012-15 phage adsorption assay. Interestingly, the phages could attach to *S. aureus*, but not to CA. After 1 min, more than half of the phages were absorbed, but over 30% of the phages remained free after 20 min (Figure 4b).

Unlike other phages specific to aerobic bacteria, which are measured for shorter periods of time, the one-step growth curve of the 2012-15 phage was measured for a longer period (18 h). The results from this measurement showed that the latent period (the time between absorption and the start of the first burst) was 6 h. The first round of infection was terminated 14 h post-infection and the burst size was determined to be approximately 2700 plaque-forming units (PFUs) per cell (Figure 4c).

### 2.6. Phage 2012-15 Stabilities at Different Temperatures and pHs

Populations of the 2012-15 phage were exposed to temperatures of 45, 55, or 65 °C for 1 h. Approximately 15% of the phages were inactivated at 45 °C, but 90% of the phages were inactivated at 55 °C. At 65 °C, the phages were completely inactivated (Appendix A).

After stability tests at various hydrogen ion concentrations, the phages were found to be stable at pH values ranging from 4 to 8 for 24 h, but over 60% and 80% of the phages were inactivated at pH values of 9 and 10, respectively (Appendix A).

## 3. Discussion

CA is a major bacterial component of the human skin microbiome [23]. With advances in molecular subtyping technology, CA can be subdivided and differentiated pathogenically [24]. However, the pathogenicity of the various subtypes of CA is still under investigation. Modern microbiological approaches to building and maintaining healthy microbiomes can be used to eradicate harmful bacteria without disturbing beneficial bacteria. Therefore, phage applications have tremendous potential for use in maintaining or regulating a healthy microbiome and as an alternative to traditional antibiotics [25,26].

CA phages have been isolated in limited numbers since their first isolation by Brzin [27] in 1964 to date and those isolates are all *Siphoviridae* families [28]. In the present study, we isolated 15 new CA phages and characterized three isolates for their phenotypes and genotypes. Most CA clinical isolates were susceptible to those phages; however, clinical isolates belonging to ST53 (the 113 and 2878 isolates) were lysed only by the P1, P10, and P14 phages. The ST53 clone (phylogroup IB) was isolated from medical equipment and soft tissues and is known to lack the CRISPR-Cas locus [29,30]. The reason why the ST53 clone was not susceptible to most of the isolated phages should be investigated in the future.

The three sequenced phages (2012-15, P1, and P2) were genomically similar, which is consistent with previous reports, i.e., the phage genomes had similar sizes (approximately 30 kb), numbers of ORFs (~44), and ORF arrangements [31]. Phylogenetic analysis of 96 CA phages showed that the 2012-15, P1, and P2 phages were closely related to other reported phages, and MAUVE analysis with even the distantly related Kubed phage indicated that genetic rearrangements did not occur among them. The limited diversity in phages suggests that the interaction between CA and phages on human skin is not competitive.

Based on a hydrophobicity/hydrophilicity analysis of the phage proteins, the capsid (major head) proteins and TMPs of phages contained more than 50% hydrophobic AAs and a lower number of charges than *E. coli*, *Bacillus*, *Lactococcus*, *Listeria*, and *Streptococcus* phages. Proteins containing 50–75% hydrophobic residues may be insoluble or only partially soluble in aqueous solutions, even if the protein contains 25% or more charged AAs [32]. The capsid protein is a major structural protein that makes up the phage head and occupies a large volume in the phage. The TMP forms a channel structure through the host cell membrane, which the phage can use to inject its genetic information into the host cell [33].

The results of the hydrophobicity analysis explain why aggregated phages were easily observed in TEM analysis and why serial dilutions of the phages in SM buffer failed. After the isolation and propagation of phages, an attempt was made to calculate the phage titer, but the serial dilution of the phage solution with SM buffer failed; hence, it was difficult to accurately measure the phage titer. Previous studies on CA phages may have failed to report quantitative data (e.g., adsorption analysis and burst size) because of phage solubility issues. Based on the results of the hydrophobicity analysis, a mild detergent Tween 20 was added to the SM buffer to completely dissolve the isolated phages. SMT^2^, which included 2% Tween 20 in the SM buffer, produced consistent values when it was used for the dilution and titration of the phages (data not shown).

CA and its phages exist in lipid-rich environments in the skin, such as the sebum secreted from the sebaceous glands in hair follicles [34]. The habitats of both are limited to the human skin and associated tissues. Thus, phages might have evolved to be capable of adsorbing to and infecting CA in lipid-rich environments. The hydrophobic nature of phages and CA is important for the lives of both and warrants further investigation.

In the SMT^2^ buffer, over 80% of the 2012-15 phages adsorbed to CA in 1 min, and the first round of infection was terminated in approximately 14 h with approximately 2700 PFU in burst size. The growth rate of CA is slower under anaerobic culture conditions than that of *E. coli* under aerobic conditions and phage propagation is dependent on the host growth rate. The burst size of the 2012-15 phage was relatively larger than that of other enteric phages. The large halo sizes produced by the phages also reflect a large burst size.

Interestingly, the 2012-15 phage adsorbed to *S. aureus*, even though the bacterium is not a typical host, although the adsorption efficiency was lower than that for CA. *S. aureus* is one of the major commensals in the human skin [35] and studying it could lead to a better understanding of CA phages. For example, excess *S. aureus* on the skin can trap CA phages present on the cell surface, resulting in an imbalance between CA and phages, leading to acne vulgaris and exacerbating the disease.

Although the associations between strain-level differences in CA and health and disease remain to be elucidated, CA is known to be a pathogen that causes acne vulgaris and other diseases. In a previous study, we demonstrated that the 2012-15 phage showed efficacy in treating a mouse model of CA-induced acne using clinical, histological, and immunohistochemical approaches [36]. In this study, novel phages that also target CA and that are closely related to previous phages, were isolated and characterized. The hydrophobic properties of CA phages and phage characteristics in the host adsorption and growth curves were also established and presented for the first time. These novel phages can be used as resources for phage therapy and for studying the interactions between bacteria and phages in humans.

## 4. Materials and Methods

### 4.1. Phage Isolation

In total, 64 clinical samples were obtained from patients with acne vulgaris between 2012 and 2014 at Kyungpook National University Hospital, Daegu, Republic of Korea. The ATCC 11827 strain was used to isolate and amplify phages specific to CA. CA ATCC 11827 was grown in Brain Heart Infusion (BHI, Becton-Dickinson, Oakville, ON, Canada) agar media under anaerobic conditions, which were maintained using a pouch system (GasPak^TM^ EZ Anaerobe Pouch System with Indicator, Becton Dickenson, MD, USA) at 37 °C for 72 h. Then, the bacteria were harvested and resuspended in 0.5 mL fresh BHI broth until the optical density at 600 nm was approximately 1.5. This bacterial solution was then added to 4.5 mL BHI broth containing the sample (0.2–0.5 g) and incubated under static anaerobic conditions at 37 °C for 3 to 5 days. After incubation, 0.5 mL chloroform was added and the mixture was vortexed vigorously and stored overnight at 4 °C. The solution was centrifuged at 10,000 rpm at 4 °C for 10 min and the supernatant was harvested and filtered using a 0.45 µm syringe filter. Ten microliters of the filtered solution was spotted onto a CA lawn on a BHI agar plate and incubated anaerobically to observe the clear regions generated as a result of bacteriophages lysing their host cells. The double-layer agar method was used to isolate the bacteriophages. Then, 100 µL of the filtered solution and 100 µL of the CA ATCC 11827 bacterial solution were added to 10 mL warm BHI agar solution (0.7%), mixed gently, poured onto a BHI agar plate and the mixture was allowed to solidify. Subsequent experiments, such as the incubation and amplification of phages, were performed as described previously [37]. The CA phages were resuspended and diluted in the SMT^2^ buffer (SM buffer (100 mM NaCl, 8.1 mM MgSO_4_·7H_2_O, 50 mM Tris·Cl pH 7.5, 0.01% gelatin) containing 2% Tween 20) and stored in the SMT^2^ buffer containing 20% glycerol at −70 °C.

### 4.2. Bacterial Culture

CA was cultured at 37 °C on blood agar plates with 5% sheep blood (BAP) or in BHI broth under anaerobic conditions for 3–4 days. Anaerobic conditions were maintained using either a GasPak EZ anaerobic pouch system or a Bactron anaerobic chamber (Sheldon Manufacturing Inc., Cornelius, OR, USA) connected to anaerobic gas (90% N_2_, 5% CO_2_, 5% H_2_).

### 4.3. Determination of Bacterial Host Range

For the host range determination, 18 CA clinical isolates with eight different multi-locus sequence types (MLSTs), six isolates of *S. aureus*, six isolates of *S. epidermidis*, two isolates each of *Corynebacterium* and *Bacillus*, six isolates of *P. aeruginosa*, three isolates of *A. baumannii*, five isolates of *Enterococcus faecium*, and two isolates of *Escherichia coli* that were collected and obtained from Kyungpook National University (KNU) Hospital and Korean National Culture Collection for Pathogens (NCCP) were used in the phage spot test.

### 4.4. Transmission Electron Microscope Analysis

For morphological analysis, phage particles were deposited on carbon-coated copper grids. Then they were negatively stained with 2% uranyl acetate and viewed via TEM (Hitachi-7000 operated at 60 kV; Tokyo, Japan).

### 4.5. Genome Sequencing

Genomic DNA from the three phages was purified using a Qiagen Lambda kit (Qiagen, Hilden, Germany), using the manufacturer’s instructions. Purified genomic DNA was fragmented by physical shearing. The fragments were blunt-end repaired and dephosphorylated for ligation with the pCB31 vector (Macrogen, Seoul, Korea) to create a shotgun library. The ligated product was introduced into competent *E. coli* DH10B cells and the recovered plasmids were subjected to sequencing (ABI Prism 3730xl instrument; Applied Biosystems, Foster City, CA, USA) with more than 20-fold coverage of the phage genome. Genes, tRNA, and rRNA were identified using the Glimmer, tRNA-Scan, and HMMER programs with EzTaxon-e rRNA profiles [38,39,40,41]. The identified genes were annotated using RefSeq, catFam, COG, and SEED [42,43,44,45]. The genome sequences of phage 2012-15, P1, and P2 were deposited in GenBank (accession numbers: KJ722067.1, KY926792.1, and KY926793.1, respectively).

### 4.6. Bioinformatic Analysis

Ninety-six CA phage genomes, including those of the 2012-15, P1, and P2 phages, were analyzed using BLAST Tree View as a slanted cladogram to generate a phylogenetic tree (https://www.ncbi.nlm.nih.gov/blast/treeview/treeView.cgi (accessed on 5 February 2022)). To analyze multiple conserved genomic sequence alignments with rearrangements, MAUVE version 20150226 was used [46]. The NCBI genome comparison tool and constraint-based multiple alignment (COBALT) tool were used for three different endolysin amino acid sequence alignments [47]. The hydrophobicity/hydrophilicity of the TMPs of various *Siphoviridae* phages were calculated using a web-based program (https://peptide2.com/N_peptide_hydrophobicity_hydrophilicity.php (accessed on 25 February 2022)).

### 4.7. Adsorption Assay and One-Step Growth Curve

The adsorption assay and one-step growth experiment were performed as described previously [37,48] with minor modifications. The CA ATCC 11827 strain used for the initial phage isolation was grown in BHI medium. The host cells (optical density at 600 nm, OD_600_ = 0.5) were infected with the phage suspension to a multiplicity of infection (MOI) of 0.01 in the SMT^2^ buffer and incubated at 37 °C. Samples were collected at 0, 1, 3, 10, and 20 min after infection and immediately filtered through a 0.45 μm pore size syringe filter. The titers of the unadsorbed phages were measured using the double-layer agar plate method, as described above. SMT^2^ buffer containing the initial amount of phage but without host bacteria was used as a control. The phage adsorption rate was calculated as follows: [(control titer − residual titer)/control titer] × 100.

A one-step growth experiment was performed to determine the burst size of the 2012-15 phage. Exponentially growing CA ATCC 11827 (OD_600_ = 0.5) was infected with the 2012-15 phage at a MOI of 0.0001 in BHI broth and allowed to adsorb at 4 °C for 60 min. The mixture was centrifuged at 12,000 rpm for 5 min and the bacterial pellet was resuspended in 10 mL of fresh BHI broth, followed by further culture at 37 °C under anaerobic conditions. Samples were collected every 2 h for up to 18 h and immediately titrated using the double-layer agar plate method. All assays were independently performed in triplicate.

### 4.8. Temperature and pH Stability Test

For the temperature stability tests, phage populations were exposed to temperatures of 45, 55, or 65 °C for 1 h, and phage titers were measured and compared with a control sample that had been exposed to a temperature of 35 °C for 1 h.

To investigate the pH stability of the phage, 2012-15 citrate-phosphate buffer was prepared at different pH values (from 4 to 10) and the experiments were performed as reported previously [49]. After incubation in buffers of different pH values for 24 h, a plaque assay was performed as described above.

## 5. Conclusions

We isolated 15 phages specific to CA from 64 clinical samples obtained from acne vulgaris patients. Three of these phages were selected according to host spectrum results and characterized genomically and phenotypically. We discovered, for the first time, the hydrophobic properties of those phages and proposed a detergent-containing buffer for studies with CA-specific phages.

## Figures and Tables

**Figure 1 antibiotics-11-01041-f001:**
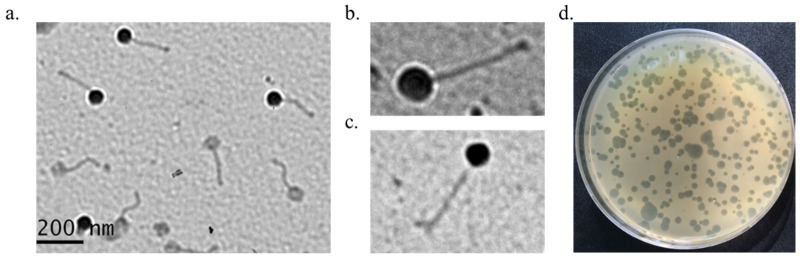
Transmission electron microscopy (TEM) images and plaque morphology of the isolated phages. (**a**) 2012-15 phage, (**b**) P1 phage, (**c**) P2 phage, (**d**) plaques of 2012-15 phage.

**Figure 2 antibiotics-11-01041-f002:**
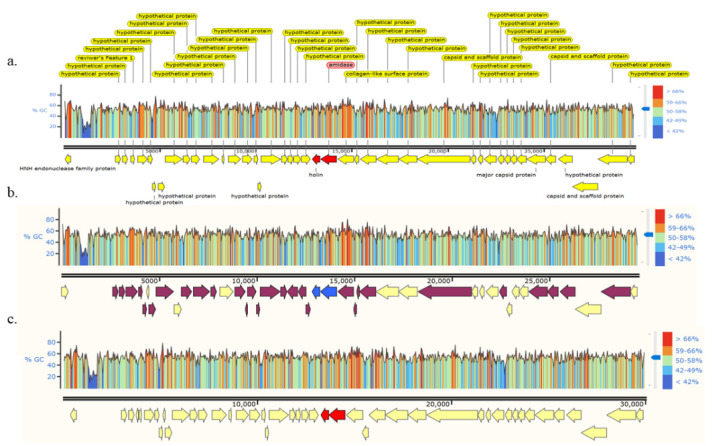
Genomes and open reading frames (ORFs) of three isolated phages, i.e., 2012-15 (**a**), p2 (**b**), and P2 (**c**). Identical ORFs across the phages (yellow arrow); those of the P1 phage were not identical with those of the 2012-15 and P2 phages (maroon arrows); holing (small blue and red arrows) and endolysin (large blue and red arrows) are shown. The arrangement of the ORFs was nearly identical across all three phages, as were the GC contents of each genome (*Y* axis).

**Figure 3 antibiotics-11-01041-f003:**
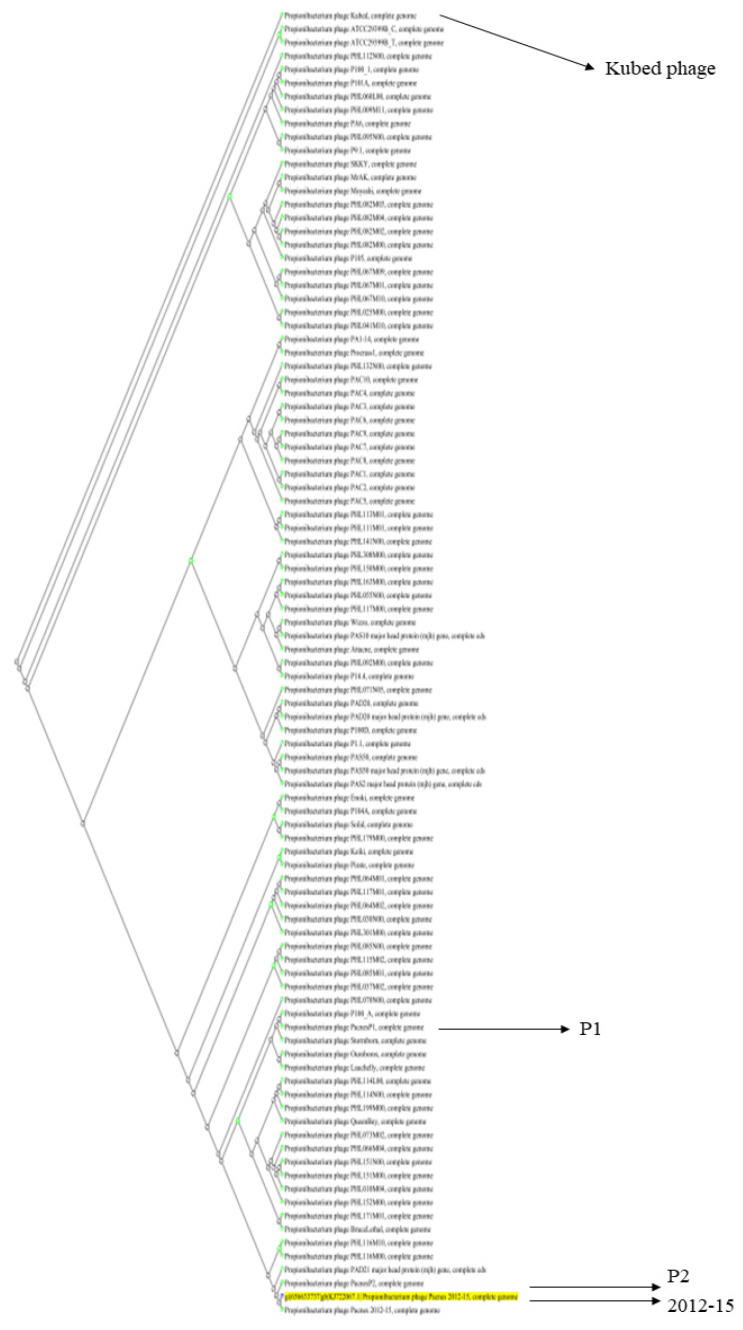
Phylogenetic slanted cladogram of 96 *C. acnes* (*Propionibacterium acnes*) bacteriophages. The 2012-15, P1, and P2 phages are marked with arrows. The Kubed phage is the farthest one from the 2012-15 and P2 phages.

**Figure 4 antibiotics-11-01041-f004:**
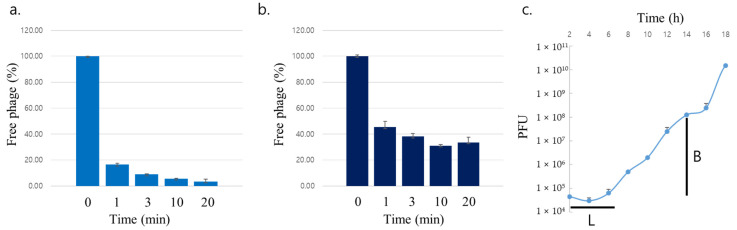
Adsorption assays of 2012-15 phage using *Cutibacterium acnes* ATCC 11827 (**a**) and *Staphylococcus aureus* ATCC 25923 (**b**). One-step growth curve of 2012-15 phage with *Cutibacterium acnes* ATCC 11827 (**c**). L stands for the latent period and B for the burst size of the phage.

**Table 1 antibiotics-11-01041-t001:** Host range determination of *C. acnes* bacteriophages using the clinical isolates of *C. acnes*.

Phage	Multi-Locus Sequence Type of *C. acnes* Clinical Isolates															
	ST1		ST2		ST6			ST22		ST25	ST53		ST69			ST115
	109	893	2874	105	2875	1490	1912	114	106	2877	113	2878	391	2876	1475	112
2012-15	+	+	+	+	+	+	+	+	+	+	−	−	+	+	+	+
P1	+	+	+	+	+	+	+	+	+	+	+	+	+	+	+	+
P2	+	+	+	+	+	+	+	+	+	+	−	−	+	+	+	+
P3	+	+	+	+	+	+	+	+	+	+	−	−	+	+	+	+
P4	+	+	+	+	+	+	+	+	+	+	−	+	+	+	+	+
P5	+	+	+	+	+	+	+	+	+	+	−	+	+	+	+	+
P6	+	+	+	+	+	+	+	+	+	+	−	−	+	+	+	+
P7	+	+	+	+	+	+	+	+	+	+	−	−	+	+	+	+
P8	+	+	+	+	+	+	+	+	+	+	−	−	+	+	−	+
P9	+	+	+	+	+	+	+	+	+	+	−	−	+	+	+	+
P10	+	+	+	+	+	+	+	+	+	+	+	+	+	+	+	+
P11	+	+	+	+	−	+	+	+	+	+	−	−	+	+	+	+
P12	+	+	+	+	−	+	+	+	+	+	−	−	+	+	+	+
P13	+	+	+	+	−	+	+	+	+	+	−	−	+	+	+	+
P14	+	+	+	+	+	+	+	+	+	+	+	+	+	+	+	+

**Table 2 antibiotics-11-01041-t002:** Hydrophobicity of major structural proteins of 2012-15 and ADB-2 bacteriophages. Bold numbers indicate proteins with more than 50% hydrophobic amino acid composition and which may be insoluble or partially soluble in aqueous solutions.

Phage	Structural Protein (Number of Amino Acids)	Acidic Amino Acids (%)	Basic Amino Acids (%)	Neutral Amino Acids (%)	Hydrophobic Amino Acids (%)
2012-15 phage	Tape measure protein (921)	**3.58**	**8.9**	**31.81**	**55.7**
	Capsid protein (323)	**10.84**	**9.91**	**28.48**	**50.77**
	Minor tail protein (280)	11.07	13.21	39.64	36.07
	Minor tail protein (87)	10.34	12.64	32.18	44.83
	Minor tail protein (272)	9.93	9.19	34.93	45.96
	Minor tail subunit (315)	14.6	10.48	29.84	45.08
	Major tail protein (210)	14.29	12.38	32.38	40.95
	Head scaffold protein (183)	21.31	16.94	28.42	33.33
	Portal protein (441)	14.06	10.66	30.61	44.67
*Escherichia coli* ADB-2 phage (NC_019725)	Tape measure protein	13.06	12.75	33.54	40.65
	Capsid protein	17.03	15.14	25.14	42.7
	Minor tail protein	9.4	8.55	42.74	39.32
	Tail component 1	12.93	9.64	38.19	39.25
	Tail component 2	7.04	10.05	37.69	45.23
	Tail fiber	14.52	11.45	33.87	40.16

**Table 3 antibiotics-11-01041-t003:** Hydrophobicity of the tape measure proteins from 12 *Siphoviridae* bacteriophages. Bold numbers indicate proteins with more than 50% hydrophobic amino acid composition, which may be insoluble or partially soluble in aqueous solutions. AAs: amino acids.

Tape Measure Protein of Bacteriophage (Number of Amino Acids)	Acidic Amino Acids (%)	Basic Amino Acids (%)	Neutral Amino Acids (%)	Hydrophobic Amino Acids (%)	Accession Number
*Cutibacterium acnes* phage 2012-15 (921)	**3.58**	**8.9**	**31.81**	**55.7**	YP_009151455.1
*Escherichia coli* phage Lambda (853)	11.84	13.25	35.05	39.86	NP_040595.1
*Escherichia coli* phage T5 (1226)	12.23	11.83	36.62	39.31	VUF55695.1
*Bacillus* phage SPP1 (1032)	**4.94**	**12.79**	**31.88**	**50.39**	Q0PDK7.1
*Lactococcus* phage c2 (624)	8.97	12.34	41.51	37.18	ASZ70817.1
*Lactococcus* phage P2 (999)	7.21	7.71	39.74	45.35	D3WAD2.1
*Lactococcus* phage 1358 (690 AAs)	7.39	8.84	35.65	48.12	YP_009140403.1
*Lactococcus* phage TP901-1 (937 AAs)	8.75	9.71	34.79	46.74	AAG32164.1
*Lactococcus* phage Tuc2009 (1025 AAs)	8.68	9.56	34.93	46.83	NP_108725.1
*Listeria* phage PSA (1026 AAs)	13.16	11.4	30.31	45.13	NP_510993.1
*Streptococcus* phage STP1 (1656 AAs)	11.11	15.22	35.14	38.53	ATI20023.1
*Staphylococcus* phage 80α (1154 AAs)	5.89	10.23	34.06	49.83	ABF71627.1

## Data Availability

The genome sequences of phages 2012-15, P1, and P2 have been deposited in GenBank (accession numbers: KJ722067.1, KY926792.1, and KY926793.1, respectively).

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
