# Peer review of "Genomic and Phenotypic Characterization of Cutibacterium acnes Bacteriophages Isolated from Acne Patients"

_antibiotics, 2022, doi:10.3390/antibiotics11081041_

Round 1

Reviewer 1 Report

Dear Authors,

Congratulations for the elaboration and results obtained in the manuscript, I suggest some modifications.

In the keywords I would suggest other words that are not present in the title of the manuscript.

The authors isolated, characterized (genomically and phenotypically) specific phages for C. acnes. The object of study is C. acnes, however there is an excess of this word in the manuscript, totaling 92 times. In the discussion the word C. acnes is repeated 34 times in 74 lines.

I suggest a modification in the wording to reduce the amount of the word C. acnes.

Figure 2A, supplemental material, would fit readings at temperatures between 45 and 55 degrees Celsius.

Figure 2B, supplemental material, the authors used the same buffer in a pH range between 4-10. Would it be ideal to use different buffers for different pH ranges?

I suggest a topic of conclusion of the obtained results.

Author Response

Dear Reviewer 1,

Thank you for your review and comments.

Based on your suggestion for the wording, we modified C. acnes to CA for a simple and economical presentation.

In Figure 2S, the phage stability decreased sharply between 45 and 55°C, so we further investigate the phage stability using a narrower temperature interval between 45 and 55°C.

For phage stability in various pH ranges, we used a citrate-phosphate buffer. For the application of phage to human skin, pharmaceutical ingredients accepted for human skin are more suitable for research.

According to reviewer’s comment, we added Conclusion next to M&M section. Thank you very much.

Reviewer 2 Report

Comments to the author(s)

The manuscript by Kim et al describes the isolation of 15 phages (and subsequent ultrastructural and genomic characterisation of 3 phages) targeting the gram-positive bacterium Cutibacterium acnes, a commensal bacteria and causative agent of human acne among other conditions. The authors motivations for this body of work stem from the idea that phages as a part of phage therapy can be used as an alternative to antibiotics, with increasing antibiotic resistance seen in C. acnes worldwide. The authors isolate 15 phages using clinical samples of patients with acne vulgaris and these phages show relatively broad lytic ability against multiple clinical isolates. The authors also use TEM to indicate three of these phages have a morphology consistent with a siphovirus (long non-contractile tail with an isometric head). The authors then sequence and genomically characterise the three phages and show they are similar to previously published C. acnes phages. The authors also discuss the hydrophobic characteristics of the phage proteins which is an interesting discussion point given the environment the host and phages occur in. Finally, the authors perform some routine assays to determine burst size, temperature, and pH characteristics of the three phages.  

The paper is well written and reasoned, the bulk of my comments will focus on the genomic characterisation of the three phages where I think the most improvements can be made.

The relationship between the nucleotide sequences the three phages could be made clearer to the reader. While you do illustrate similarities by making note of similar phage sizes, GC content, and then the phylogenetic tree, I think a simple average nucleotide identity (ANI) or average amino acid identity (AAI) comparison between the three phages could very quickly illustrate to the reader that 2012-15 and P2 are essentially the same phage (99.8% ANI through a quick investigation on my part), while P1 is quite different (~85% ANI to the other two phages). Tools you can use for this type of analysis (web-based and very easy, performed in less than a minute) are VIRIDIC (more of a phage-specific ANI tool) accessed via http://rhea.icbm.uni-oldenburg.de/VIRIDIC/  or JSpeciesWS (generalised ANI tool) accessed via https://jspecies.ribohost.com/jspeciesws/. If you wish to take it a step further, you could perform VIRIDIC using a larger sample of the C. acnes phages from the NCBI database and illustrate phage groupings based on similarity.

Improvements can be made to the gene identification in the three phages, my analysis of the genomes has brought up genes that I do not believe are actual genes, and some genes that are likely longer/shorter than currently annotated. I will make a list here and trust you will double check these genes before resubmission 

2012-15 phage:

·      Hypothetical protein. Location: 3416-3568

o   Likely not a gene. Lacks an obvious ribosomal binding site (RBS), is running in opposite orientation compared to neighbouring genes, and is overlapping a gene extension I propose in the next point.

·      hypothetical protein. Location: 3588-3737

o   I believe this gene actually begins at 3486 where there is a more appropriate RBS.

·      Hypothetical protein. Location: 4809-4940

o   Likely not a gene. Similar to the first example above, it lacks an obvious RBS, is in the wrong orientation, and the proposed start codon is much too close to the neighbouring gene to expect a valid promoter to drive its expression.

PacnesP1 phage:

·      Hypothetical protein. Location: 9424-9549

o   Likely not a gene. Lacks an RBS and is in the wrong orientation compared to neighbours.

·      Hypothetical protein. Location: 9554-10,009

o   I believe this gene actually begins at 9452.

·      Group of genes from ~11,500-12,700. 

o   I believe these are incorrectly identified in the current state. This region likely contains three genes in the forward direction (11,576-11,851, 11,855-12,244, and 12,249-12,764), not in the reverse. Please double check your current annotations in this region and rectify as appropriate.

·      Tail assembly chaperone (2 genes annotated, 21,025-21,312 and 21,411-21,662). 

o   This gene is notorious for containing a ribosomal slippage sequence causing a frameshift during transcription. The result of this frameshift causes two different tail assembly chaperone proteins to be produced, the more frequent being a shorter one (corresponding to 21,411-21,662 coding sequence), and a longer one where a frameshift occurs during transcription causing a combined product from both genes. This is well documented in this protein and many examples can be found online, but this page is a good example (https://cpt.tamu.edu/training-material/topics/phage-annotation-pipeline/tutorials/annotating-tmp-chaperone-frameshifts/tutorial.html). In this case, I believe the slippage occurs between 21,434-21,429 (AAAGGGG) leading to a -1 framing shift allowing expression all the way to the 21,025 generating a ~230a.a combined chaperone protein.

PacnesP2 phage:

Given the similarity to 2012-15 phage, please just double check any changes you make are equally applied to this phage annotation.

I believe improvements can also be made to gene annotations. For instance, the tape measure protein in Siphoviruses is one of the easiest genes to identify, even by eye, (is usually the largest gene in the genome), however, it has been incorrectly annotated in both 2012-15 and PAcnesP2 even though multiple methods of gene searching were purported to be employed (RefSeq, catFam, COG, and SEED). I suggest the authors have another look at these annotations. For structural proteins, the use of the Virfam webserver (http://biodev.cea.fr/virfam/) can offer more sensitivity for detection.

With a quick usage of Virfam for 2012-15 phage I was able to more annotate several more genes. 

·      Hypothetical protein. Location: 27,816-29,327 => Can be annotated as a large terminase subunit.

·      Capsid and scaffold protein. Location: 26,494-27,819 => Can be annotated as a portal protein.

·      Hypothetical protein. Location: 21,890-22,522 => Can be annotated as a major tail protein.

·      Hypothetical protein. Location: 23,597-24,076 => Can be annotated as a head completion protein.

·      Hypothetical protein. Location: 23,248-23,595 => Can be annotated as a head closure protein.

·      Hypothetical protein. Location: 22,951-23,241 => Can be annotated as a neck protein.

·      Hypothetical protein. Location: 22,583..22,915 => Can be annotated as a tail completion protein.

Author Response

Dear Reviewer 2,

Thank you for your comments about phage gene annotation.

We revised gene annotation according to your suggestion. And the information you suggested is valuable for us to study further phage genomics. Virfam system was unstable when we visited, but we could know that phage gene annotation needs curation with various sources. According to revised ORFs, we revised the main text, figure 2, and supplementary table. Thank you very much.

This manuscript is a resubmission of an earlier submission. The following is a list of the peer review reports and author responses from that submission.